# Do Intellectually Gifted Children Show Similar Physical Habits, Physical Fitness Levels and Self-Perceived Body Image Than Typically Developed Children?

**DOI:** 10.3390/children9050718

**Published:** 2022-05-13

**Authors:** Aroa Otero Rodríguez, Miguel Adriano Sánchez-Lastra, José Carlos Diz, Carlos Ayán

**Affiliations:** 1Grupo de Investigación Wellness and Movement, Departamento de Didácticas Especiais, Facultade de Ciencias da Educación e do Deporte, Universidade de Vigo, 36005 Pontevedra, Spain; misanchez@uvigo.es (M.A.S.-L.); jcdiz@uvigo.es (J.C.D.); cayan@uvigo.es (C.A.); 2Instituto de Investigación Sanitaria Galicia Sur (IIS Galicia Sur) Sergas-UVIGO, 36312 Vigo, Spain

**Keywords:** gifted children, child exceptional, sedentary behavior and physical fitness levels

## Abstract

Scientific evidence regarding whether intellectually gifted children show similar physical activity habits and physical fitness levels in comparison to typically developed children, is inconclusive. This is in part due to the scant research that has directly compared both groups of people. In this study, physical activity prevalence, self-perceived and objectively assessed physical fitness levels, and body image were assessed in a sample of 74 intellectually gifted children (mean age 11.6 ± 1.2 years). Seventy-four non-gifted children matched by age and sex were selected as a comparison cohort. Results indicated that both groups showed similar self-perceived and objectively assessed fitness levels. Physical activity habits were also similar, although significant differences were observed indicating that intellectually gifted girls were less active in comparison with non-gifted girls. Both cohorts perceived their body image accurately, although intellectually gifted children were much more satisfied with their physical appearance.

## 1. Introduction

Research on the prevalence of physical activity (PA) among children and adolescents is on the rise. Different investigations have associated PA levels with children’s quality of life and physical and mental health [1,2,3]. The World Health Organization [4] recommends children and adolescents engage in at least 60 min per day of moderate to vigorous-intensity physical activity (MVPA) over the course of a week [4]. Thus, studies are being carried out to determine whether children and adolescents are actually meeting the recommended PA guidelines [5,6]. Similar research has also been focused on assessing physical fitness (PF) levels in children and adolescents [7,8,9], as PF is considered an important marker of health even at a young age [10].

Investigations in both PA prevalence and PF levels are commonly conducted with typically developing children and/or children with physical or intellectual disabilities. Very few studies, however, have investigated PA prevalence and PF levels in children who are intellectually gifted (IG), a group that is largely ignored in this area of research. Previous research on the PA habits of IG children is mostly embedded within studies focused on extracurricular practices, [11,12,13], and those that do not follow extra-curricular practices [14,15], have limitations restricting generalizability. For example, Hormazábal-Peralta et al. [14], compared the physical characteristics of IG boys and girls, but failed to include a sample of typical development children; thus, a direct comparison cannot be made. Therefore, further research is needed.

Similarly, scant research has been conducted regarding the PF levels of IG children, and those that have been conducted, are incomplete. For example, although Çakiroğlu, [16] examined the PF of fifty IG children in comparison to typical development children [17], the researchers failed to measure cardiorespiratory fitness—an important predictor of both physical and mental health—as well as academic achievement [18]. Similarly, results obtained by Song and Ahn [17] were limited by the fact that the sample contained only IG students with aptitude in mathematics and science.

Thus, it seems that current scientific evidence is not strong enough to debunk existing myths regarding IG children, such as stereotypes portraying them as sickly, weak, and not motivated towards the practice of PA and sports [19]. More empirical evidence is therefore needed to eliminate persisting misconceptions such as the idea that IG students are not likely to be good at athletics [20].

Finally, it must be highlighted that while there is research associating social-psychological outcomes with PA and sports participation, such as global self-concept among IG children and adolescents [21], there is still a lack of research regarding how this population perceives their PF levels and body image. Obtaining data on both outcomes could help to increase the existent body of research related to well-being in this population, which is somehow inconsistent [22].

The aim of this study was to identify whether IG children and adolescents exhibit similar PA patterns and PF levels in comparison to their typical development peers. The secondary aim was to provide information on IG students’ self-perceived PF and body image.

## 2. Methods

### 2.1. Participants

In this observational cohort study, the target population included all students in grades 9 to 15 enrolled in urban schools across the autonomous community of Galicia (north of Spain). Identification procedures involved an initial telephone call to a total of 362 schools to find children or adolescents previously identified by their teachers or their parents as IG who might agree to take part in this study. In the Spanish context, due to the current educational legislation [23], each geographic region selects their own criteria for identifying IG students. In the geographic region of Galicia, Renzulli’s Three-Ring Conception of Giftedness [24,25] guides identification; that is, students are recommended for identification as IG if they show high intellectual ability, creativity, and engagement. First, parents or teachers who suspect that a student is IG will make a recommendation, and next, the school counselor will evaluate that child’s skills and abilities.

For the purpose of this research, in those schools where IG children were previously identified and willing to take part in the investigation, the main author of this research interviewed them and administered the Spanish version of the Wechsler Intelligence Scale for Children (WISC) [26] to confirm that the selected children did in fact qualify as IG. To be included in the IG cohort, participants had to score ≥130 on the full-scale WISC. As a comparison group, (i.e., typical development cohort), we selected students from the same schools in a 1:1 ratio matched by age and sex. We assumed that children in the comparison group were not previously identified as IG and that they came from a similar sociodemographic and economic background since they were attending the same urban public schools. Children were excluded from the study if they: (a) showed any medical condition that could hinder performance of the field-based tests; (b) had educational special needs or learning problems, and (c) reported issues that prevented them from performing their usual PA. Before the study began, written informed consent was obtained from the children and their parents through the use of a standard informed consent form developed by the University of Vigo. The study was conducted in accordance with the Declaration of Helsinki, and the protocol was approved by the Ethics Committee of the Faculty of Education and Sport Science, University of Vigo (Ref: 104721) on 4 July 2021.

### 2.2. Instruments

#### 2.2.1. Physical Activity

The prevalence of PA was assessed by means of the International Physical Activity Questionnaire-Short Form (IPAQ-SF). Participants self-reported their duration of walking, moderate and vigorous physical activities lasting more than ten minutes in a typical week, and the time spent sitting. Total weekly MVPA was calculated as metabolic equivalents (METs)—minutes per week from the sum of walking (3.3 METs), moderate (4 METs), and vigorous (8 METs) activities, following the IPAQ-SF protocol for data processing and analysis (The IPAQ Group, 2005). Participants were then classified as to whether or not they met current WHO PA guidelines based on achievement of 420 min per week of total PA (60 min per day × 7 days).

#### 2.2.2. Physical Fitness

For assessing body composition, the children’s weight (kg) and height (cm) were measured by means of a digital scale and a stadiometer. The body mass index (BMI) was calculated by dividing the body weight in kilograms by the height in meters squared (kg/m^2^). Cardiorespiratory fitness was measured by means of identifying heart rate (HR) response before and after performing a submaximal effort test. Children were instructed to perform a skipping exercise while alternately raising their heels to their buttocks at a rate of two touches per second over the span of one minute. HR was registered five minutes before the test began (HR basal), just after the test was finished (HR maximum), and at one-minute intervals for five minutes using an HR monitor (Polar RS400, Kempele, Finland) that was connected via Bluetooth to an iPad Air 2. We calculated HRR60 to assess HR recovery during the initial fast phase, defined as the difference between maximum HR and HR after 60 s (in beats per minute and in percentage) [27]. A handheld dynamometry test from the Eurofit battery [28] was used to assess muscular strength. Participants held a pediatric dynamometer (Tecnomed 2000) with their preferred hand (with the arm hanging down by the side of the body) and squeezed as hard as possible for at least 2 s. Flexibility was measured by means of the V sit test [29]. Children were instructed to sit on the floor with their knees straight and feet separated by about 12 inches to form a V-shape leg position. A ruler was placed between the legs with the 0-inch mark located at the heel line. The children, palms downs and placing one hand on top of the other, were told to slowly reach forward as far as possible, while at the same time, sliding their hands along the ruler and then holding that position for two seconds. For the muscular and flexibility tests, two trials were allowed, and the highest scores were registered.

#### 2.2.3. Self-Perceive Physical Fitness

Self-reported PF levels were identified through the administration of the International Fitness Scale (IFIS) [30]. The IFIS is comprised of 5 Likert scale questions about self-reported fitness (very poor, poor, average, good, and very good) relating to perceived overall fitness and its main components: cardiorespiratory fitness, muscular strength, speed and agility, and flexibility.

#### 2.2.4. Body Perception

The Stunkard pictogram adapted for young people, which consists of 7 silhouette figures that increase gradually in size from very thin (a value of 1) to very obese (a value of 7), was used to assess body perception [31]. The participants were first asked to choose the figure that reflected how they thought they looked. Afterward, they were asked to choose their ideal figure. Finally, the participants indicated how they would like to be seen by others. A body size satisfaction variable was created by subtracting the number of the figure selected as the ideal body size from the number of the figure selected as the self-body size. This satisfaction variable had three categories: if they wished to have more weight, to remain the same weight, or to have less weight.

The measurements were performed by a specialist in psych pedagogy with previous experience in fitness testing. Children who are IG were individually interviewed and assessed in their school, while typical development children answered the questionnaires and performed the PF tests in groups during the physical education lessons.

### 2.3. Procedure

We first contacted schools to find IG students. Once the schools agreed to participate in the study, we informed the families about the study. Finally, we presented the study in the classroom, and students who agreed to participate and had their parents’ permission were included in the study. Participants completed the study in physical education class. First, they provided demographic information and completed three questionnaires about physical activity, physical fitness, and body perception. Next, all participants were measured, weighed, checked for flexibility, heart rate, and strength, and these data were recorded by the researchers. This same procedure was followed for non-gifted participants.

### 2.4. Data Analysis

We assessed normality by means of the Kolmogorov–Smirnov test. Data are presented as mean ± standard deviation (SD) for normally distributed variables, and median (interquartile range, IQR) if they did not show a normal distribution. Qualitative variables are presented as n (%). We compared quantitative variables with Mann–Whitney U test or *t*-Student test, and qualitative variables with Chi-squared test, as appropriate. A two-sided P value of less than 0.05 indicated statistical significance. All the analyses were performed with the IBM SPSS Statistics for Macintosh, version 28.0 software [32].

## 3. Results

Out of the 362 schools initially contacted, 18 refused to take part in the investigation, and 322 informed us that no IG children had been identified among their students. Thus, a total of 40 schools were finally selected for recruiting IG children and a comparison group.

We included 148 students in the study, with an average age of 11.6 ± 1.2 years (median 12 years, range 9–15), 78 male (11.9 ± 1.2 years) and 70 female (age 11.3 ± 1.1 years), with 74 in every cohort, with similar age and sex distribution. There were no differences between cohorts at baseline regarding weight (47.8 ± 12.8 kg), height (1.54 ± 0.1 m), or BMI (19.9 ± 3.53 kg/m^2^). The prevalence of PA and sedentary behavior (sitting time) is shown in Table 1.

Participants who are IG showed significantly lower values than the typical development children regarding total walking, moderate and vigorous-intensity activities per week, and weekly total PA (all *p* < 0.05). When the analysis was stratified by sex, it was shown that these differences were accentuated and only present in girls. No significant differences were shown regarding the meeting of current WHO PA guidelines or sedentary time between both groups, neither in the total sample nor stratified by sex.

Table 2 shows the comparison of fitness measurements for both groups.

No significant differences were found between both groups in any of the fitness components assessed, or for the self-perceived PF levels (Table 3).

Differences found between genders were small and of low magnitude.

Significant correlations were found between BMI and the Stunkard scale (R Pearson = 0.684, *p* < 0.001), indicating that participants perceived their own body image accurately.

In general, around 62% of the IG participants were satisfied with their body image, while only 35% of typical development participants shared this opinion. These differences in body satisfaction were striking in the case of the girls (68.5% vs. 28.5%) (Figure 1).

## 4. Discussion

This study aimed to compare PA prevalence, fitness levels, and body image between IG and typically developed children. We found that IG girls were less active than typically developed girls, while no differences were observed regarding self-perceived and objectively assessed PF between groups. The obtained results also indicated that IG children were much more satisfied with their physical appearance than typically developed children.

Sports involvement has been found to be the most popular extracurricular activity among IG children [13]. Moreover, previous research on the PA habits of IG children indicated that they tend to engage in sports activities at a rate that is at least on a par with typically developed children, if not higher [33]. Our investigation provided a deeper comparative analysis of PA habits and yielded interesting results. For instance, regarding PA patterns of both IG and typically developed children, we found significant differences in the total sample. The sex-stratified analysis showed that these differences were present in girls but not in boys. About 45% of the IG girls reported PA levels under the WHO recommendations, compared to 22% of the typical development girls, 26% of the IG boys, and 33% of the typical development boys. This is an interesting and important finding, given the lack of investigations comparing PA levels of IG with typical development children. In this regard, a very recent study found no significant differences in PA levels between IG and typically developed children, even when controlling for sex [15]. In this investigation, it was found that 33.8% of the IG and 33.5% of the typical development participants met the WHO’s recommended daily PA guidelines.

Findings observed in the IG sample are in accordance with those from Hormazábal-Peralta et al. [14], who reported that IG boys had higher PA levels than IG girls, with 20.6% of the boys and 38.46% of the girls reporting less than 2 h of PA per week. We expand these findings by including the total levels of PA considering not only frequency and quantity but also intensity, and their accordance with international recommendations. Due to the importance of meeting adequate PA levels for health outcomes, it should be further investigated why these girls tend to be less active and search for strategies to promote their PA.

Our results for PF levels indicate that there were no significant differences between IG and typical development children in flexibility, strength, cardiorespiratory fitness, and body composition. Similar results were observed by Song and Ahn [17] when comparing a sample of students who were IG in mathematics and science with a group of typical development children. However, IG children showed a significantly lower body fat percentage and higher muscle mass than typical development children. This could be due to the fact that IG children were attending a science academy in a different school system that included a specific sports and PA program, while the typical development children were recruited from ordinary schools. In the study by Çakiroğlu [16], significant differences in favor of the IG group for hand grip strength were reported; however, it should be noted that only boys were included in the sample. Given the influence of sex on children’s grip strength [34], it is plausible that the difference in the results is explained by the fact that we also included girls in the analysis.

Mean BMI values found in our study for IG children were lower than those found by Pouresmali et al. [35] and by Hormazábal-Peralta et al. [14], who also reported a low prevalence of overweight and obesity in this population. We did not find significant differences between both groups regarding BMI, but we did note that more than 30% of the measured children were either overweight (20–27%) or obese (10–17%). These findings are in accordance with the current situation, in which a worrying increase in the number of European countries with a high prevalence of overweight (over 30%) and obesity (over 10%) has been reported [36]. It is worth mentioning that the prevalence of both overweight and obesity were lower among IG children, but still, high BMI values were presented among some of them.

Self-perceived PF is a key dimension of physical self-concept [37]. The self-concept construct is central to psychological well-being because people who feel good about themselves and their abilities are likely to be more effective than individuals with low self-concept who do not feel good about themselves [38]. Our results indicated that both IG and typical development children perceived their own PF levels in a similar way. These results are in accordance with previous findings indicating that few differences exist regarding physical self-concept between both groups of people [39]. Notably, differences between genders were small, which somehow contradicts past results on IG children showing that males are more likely to have higher physical ability self-concepts than females [40]. It should be noted, however, that previous investigations focused on athletic competence yielded interesting and contrary results to those shown here. For instance, Song and Ahn [17] found that IG children had significantly lower self-perceived muscular strength and a higher self-perceived cardiorespiratory fitness in comparison with typical development children. In the same way, Infantes-Paniagua et al. [15] observed that IG children had lower self-perceived athletic competence than typical development children. These differences could be due to the fact that in our investigation IG and typical development children were attending the same schools, while this was not the case in the aforementioned studies.

We also gathered information regarding body image, which is another dimension strongly related to self-concept. Interestingly, we found that both groups of children perceived their body image accurately, but while almost one-third of IG children were satisfied with their body image, around one-third of typical development children were not. These findings show that IG children have a more positive self-concept regarding body image, which could help to explain why they also have a higher overall self-concept, as previously reported [41]. In addition, this finding indicates that body dissatisfaction is more frequent among typical development children [42]

We also gathered information regarding body image, which is another dimension strongly related to self-concept. Interestingly, we found that both groups of children perceived their body image accurately, but while almost one-third of IG children were satisfied with their body image, around one-third of typical development children were not. These findings show that IG children have a more positive self-concept regarding body image, which could help to explain why they also had a higher overall self-concept as previously reported [41]. In addition, this finding indicates that body dissatisfaction is more frequent among typical development children [42].

To the very best of the authors’ knowledge, this is the very first study in which a number of variables related to PA, PF, and self-perception of IG children are directly compared to those of the general population. In spite of the originality of its design, there are several limitations that should be taken into account when interpreting the reported findings. For instance, participants were recruited from schools in the north of Spain, and may not be representative of academically talented students in general. In addition, PA was assessed by means of a questionnaire, which is always subject to recall bias. It should also be mentioned that there were differences in the test administration protocol between the two groups. Thirdly, we assumed that children in the comparison group were not IG based on the opinion of the schools’ management, so the WISC was not administered to these students. Similarly, we assumed that both groups shared similar sociodemographic and economic backgrounds since they were attending the same schools. Subsequently, we did not collect this information or information on the families’ cultural backgrounds or the parents’ PA levels. The omission of this information limits the comparison carried out in this research. Finally, although the sample size was not small for a study of this kind, having included a greater number of IG children would have allowed the performance of a deeper analysis regarding sex influence in some of the variables assessed. Notwithstanding the limitations, this study makes an original contribution to the literature on IG children. Future studies should include a greater sample made up of IG and typical development children recruited from the same schools and cover a wider geographical location. The WISC should be administered to all participants while power and sample size estimation should be carried out beforehand. Information on the sociocultural and demographic background of all the children should be obtained, as well as their parents’ PA levels.

Children and adolescents who are IG generally exhibit similar PA patterns and PF levels to their typical development peers. However, significant differences were observed indicating that IG girls are less active in comparison with typical development girls. Both IG and typical development children perceive their PF levels and body image accurately. Nevertheless, IG children are much more satisfied with their physical appearance.

## Figures and Tables

**Figure 1 children-09-00718-f001:**
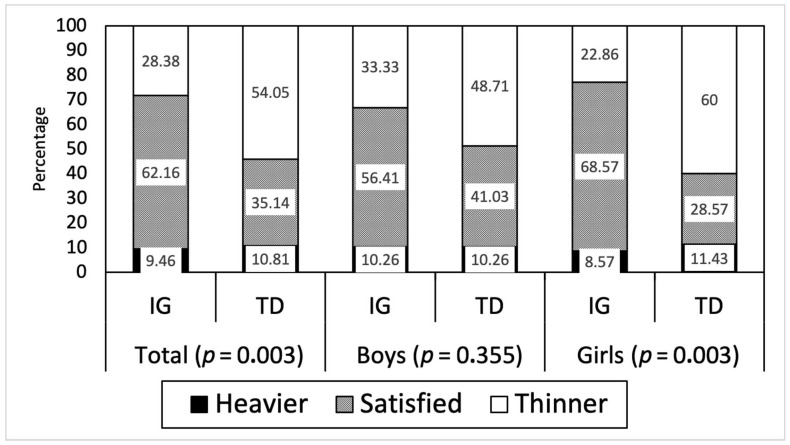
Desirable body shape, estimated as the difference between actual and desired body shape. IG: intellectually gifted; TD; typical development.

**Table 1 children-09-00718-t001:** Comparison of the physical activity and sedentary behavior levels of the sample.

IPAQ-SF Items	Total Sample (*n* = 148)			Girls (*n* = 70)			Boys (*n* = 78)		
	IG (*n* = 74)	TD (*n* = 74)	*p*	IG (*n* = 35)	TD (*n* = 35)	*p*	IG (*n* = 39)	TD (*n* = 39)	*p*
Sitting time (hours/day)	7.5 ± 1.2	7.1 ± 1.2	0.095	7.6 ± 1.1	7.1 ± 1.3	0.141	7.4 ± 1.3	7.1 ± 1.6	0.326
Walking									
Frequency (times/week)	4.6 ± 2.3	4.8 ± 2.4	0.649	4.4 ± 2.3	4.6 ± 2.5	0.693	4.8 ± 2.3	4.9 ± 2.3	0.806
Duration (minutes/time)	41.2 ± 31.9	56.4 ± 43.6	0.016	38.6 ± 25.9	52.6 ± 40.4	0.089	43.5 ± 36.6	59.9 ± 46.6	0.089
Total (METs-minutes/week)	714.9 ± 700.9	1027.0 ± 1004.8	0.030	637.8 ± 531.1	977.7 ± 904.0	0.059	784.0 ± 825.3	1071.2 ± 1097.3	0.195
Moderate intensity activities									
Frequency (times/week)	2.6 ± 1.5	2.6 ± 1.4	0.954	2.2 ± 1.2	2.8 ± 1.4	0.033	3.0 ± 1.6	2.4 ± 1.4	0.073
Duration (minutes/time)	55.1 ± 24.3	56.6 ± 40.9	0.788	56.7 ± 25.8	52.3 ± 35.3	0.551	53.7 ± 23.2	60.5 ± 45.4	0.407
Total (METs-minutes/week)	546.2 ± 423.9	663.8 ± 885.5	0.305	449.7 ± 244.7	641.1 ± 769.3	0.165	632.8 ± 524.8	684.1 ± 987.9	0.775
Vigorous-intensity activities									
Frequency (times/week)	2.2 ± 1.8	2.7 ± 1.9	0.070	1.7 ± 1.6	2.7 ± 1.8	0.031	2.6 ± 1.9	2.8 ± 2.0	0.598
Duration (minutes/time)	56.9 ± 38.9	73.7 ± 47.1	0.020	48.7 ± 37.4	74.3 ± 46.7	0.013	64.4 ± 39.2	73.1 ± 48.0	0.382
Total (METs-minutes/week)	1344.9 ± 1242.7	2043.78 ± 1921.7	0.010	987.4 ± 1039.8	2145.1 ± 2036.2	0.004	1665.6 ± 1332.7	1952.8 ± 1834.8	0.431
Weekly total MVPA									
Minutes/week	521.3 ± 337.8	729.4 ± 471.5	0.002	429.1 ± 239.5	717.9 ± 471.1	0.002	604.0 ± 391.2	739.7 ± 477.9	0.174
Total (METs-minutes/week)	2605.93 ± 1679.0	3734.6 ± 2548.4	0.002	2075.0 ± 1300.2	3764.0 ± 2694.9	0.001	3082.4 ± 1846.7	3708.2 ± 2444.7	0.206
Meeting WHO recommendations (%)	60.8	75.7	0.053	54.3	77.1	0.045	74.36	66.7	0.463

Values are mean ± standard deviation unless stated otherwise. IG: intellectually gifted; IPAQ-SF: International Physical Activity Questionnaire-Short Form; METs: metabolic equivalents; MVPA: Moderate-to-Vigorous Physical Activity; TD: Typical development.

**Table 2 children-09-00718-t002:** Comparison of fitness measurements.

Fitness Variable	IG			Typical Development			*p*-Value
	*n*	Mean	SD	*n*	Mean	SD	
Hand grip strength (dinamometer)	74	22.1	5.7	74	22.8	5.8	0.402
Flexibility (V-Sit and Reach)	74	4.4	9.1	74	5.9	10.0	0.336
Time to resting heart rate (s)	74	81.9	20.5	74	82.1	22.2	0.954
HRR60 (bpm)	74	35.3	12.8	74	36.5	13.8	0.585
HRR60 (%)	74	25.5	9.5	74	24.6	9.3	0.573
Body Mass Index (% per group)							
Normal weight	49	66.2%		46	62.2%		0.494
Overweight	17	23.0%		15	20.3%		
Obese	8	10.8%		13	17.6%		

bpm: beats per minute; HRR60: heart rate recovery in the first 60 s after the effort; IG: intellectually gifted.

**Table 3 children-09-00718-t003:** Comparison between intellectually gifted and typical development groups using Student’s *t*-test.

IFIS Items	IG (*n* = 74)		Typical Development (*n* = 74)		*p*-Value
	Mean	SD	Mean	SD	
General fitness	3.73	0.65	3.65	0.71	0.469
Cardiorespiratory fitness	3.45	0.97	3.54	0.86	0.531
Muscular strength	3.32	0.76	3.49	0.85	0.223
Speed and agility	3.88	0.84	3.86	0.88	0.924
Flexibility	3.19	1.29	3.05	0.92	0.464

IFIS: International fitness scale; SD: standard deviation; IG: intellectually gifted.

## Data Availability

The data availability statement is held by the authors of the article.

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
