# Peer review of "Do Intellectually Gifted Children Show Similar Physical Habits, Physical Fitness Levels and Self-Perceived Body Image Than Typically Developed Children?"

_children, 2022, doi:10.3390/children9050718_

Round 1

Reviewer 1 Report

I suggest a change of the title, remove the part "Findings from a comparative study" because it is unnessesary in the title. Please clarify how did the authors obtain informed consent from participants of this survey.Did you used questionnaire that was used in previous surveys or was it constructed for the present research? Which test was used to test dana for normality 'distribution? Please discuss the limitations of the study. Discuss the results of the present study and compare them to existing knowledge in the literature.References: More recent literature should be used.

Results: they are presented correctly and with relevant indicators for a more in-depth future analysis, however, the low N cannot be ignored, especially because the authors themselves assume that it is  problem.

Key words: MeSH indexed key words should be used. 

Please pay attention to English grammar

Reviewer 2 Report

x

Thank you for the opportunity to review the manuscript. The idea is interesting, but the paper presents important bias related to the sample included. Lack of information are also related to the family background of the sample that has not been considered. In the present form, I cannot suggest this manuscript for publication.

Please, add the abstract in the paper

Introduction

Line 35-38: this sentence can be integrated with line 31. If the Authors write that they performed an extensive literature review, I would like to read the methodology of the review, the electronic databased screened and so on. So, I suggest to remove to write “extensive literature review” and integrate it with line 31.

Methods

In the introduction, the Authors adopted IG for gifted children, please, be consistent in all the manuscript adopting always this abbreviation when referring to this population

Line 60-72: There is an important bias in the participants selection. The Authors adopted the WISC scale to evaluate gifted children, but information related to the control group are not provided. Please, provide this information. According to me, the selection of gifted children following a subjective opinion of the teacher or the parents is an important limit. The school performance cannot be associated with the intelligence of the children. The screening had to be performed with the WISC in all the schools included in the study. I understand that the test requires a lot of time and effort to be performed, but in this way the selection performed by the Authors is not accurate. Second, the Authors need to specify how and based on what they selected the control group. This information is fundamental. Information related to the family background are necessary also to understand the cultural level of the families and the level of physical activity of the parents. I’m of the opinion that more than the level of intelligence of the children, it is important to know the cultural / intelligence level of the parents.

Please, provide the exclusion criteria adopted such as people with disabilities or others.

Line 84 94: please provide the protocols adopted for the submaximal effort test, the handgrip test and the flexibility test.

Line 132: please, provide information about SPSS software.

Results

Line 137: “There are no differences”, are the Authors speaking about statistical differences after the normal distribution analysis?

In table 1, the sitting time for the IG children is 7.1 in the total sample but it is 7.6 and 7.4 for girls and boys respectively. I have no access to the data, please, double check the data to be sure about the total sample mean.

Discussion

Line 170-174: Instead to describe generally the manuscript, please, write the main findings of the study.

Please, provide also feedback also related to “future studies”

Round 2

Reviewer 2 Report

Dear Athors,

thank you for the improvements performed in the manuscript and for following my suggestions. The manuscript results improved and complete. I suggest it for publication.

Best regards